# An Innovative Framework for Quality Assurance in Logistics Packaging

**Henriett Matyi** * and **Péter Tamás**

Institute of Logistics, Faculty of Mechanical Engineering and Informatics, University of Miskolc,
3515 Miskolc-Egyetemváros, Hungary; peter.tamas@uni-miskolc.hu
* Correspondence: henriett.matyi@uni-miskolc.hu

**Abstract:** *Background*: As a result of the effort to satisfy unique customer needs, the complexity of production and service processes is constantly increasing. In this context, the requirements for packaging systems, essential for carrying out logistic tasks, are also diversifying, and various quality defects and problems are appearing more and more frequently. *Methods*: The research used an inductive method. While practical problems were being solved, the need for developing the concept of a packaging inspection framework arose, the lack of which was also supported by a systematic literature review. *Results*: During the concept's development, packaging errors found in the literature were identified and methods for detection and solution were systematized. A general framework was developed to identify and eliminate these errors. The applicability of the developed method was demonstrated through a complex case study, and its accuracy was verified. *Conclusions*: This research is important because, instead of using "island" solutions, in the future, companies will have a general framework available to them for handling all packaging-related errors according to a predefined methodology. This can reduce the time required for problem-solving and increase efficiency, which is a significant competitive factor.

**Keywords:** logistics; packaging managements system; quality assurance





## 1. Introduction

The essence of logistic packaging is that the packaging design should be emphasized together with marketing and product design in order to create the best possible solution, by taking into account main factors, such as the state of the goods, the geometric dimensions of the goods, the quantity and weight of the goods, the type of goods, the current types of goods handling equipment, the composition of the transport chain system, and the storage method [1].

Today's packaging must make compromises between packaging functions and consider the role of packaging. Packaging is not only a cost-driven central task but also a value-adding process in the logistics system. There are many factors to consider throughout the packaging process, starting from the supplier to the end customer. After gathering requirement information from different departments, packaging design needs to be carefully analyzed [2].

In the context of supply chain management, the purpose of logistics is to move and locate inventory, both internally and externally, to achieve appropriate time and space advantages at the lowest possible costs. Packaging is also necessary to contain and protect products from the environment [3].

The rise of Industry 4.0 and smart factories enables the collection of large amounts of data from manufacturing processes. Using this data, machine-learning methods can make predictions about product quality and conformity, eliminating uncertainty and reducing costs throughout the supply chain [4].

In addition to marketing, protection, and containment, packaging also allows for more efficient distribution and storage of products, and can therefore contribute to reducing both costs and transit times [5].

Three levels of packaging can be distinguished: primary, secondary, and tertiary. Primary packaging is the packaging of the product itself. Secondary packaging consists of boxes or containers containing a specific quantity of primary packaging. Tertiary packaging consists of pallets and large shipping containers for storage and warehousing. In addition to its main functions, packaging affects the supply chain in several ways, as it is linked to manufacturing, waste management, and transport [6].

The increasing integration between different logistics units and other internal operations in the automotive industry is increasing the importance of packaging management for the strategic logistics operations of car manufacturers [7].

In the context of a logistics system, the identification of quality defects in logistics packaging and their proper handling has a significant impact on a company's competitiveness as it can reduce losses, such as product damage, unnecessary material handling, etc.

This paper presents a systematic literature review in this area, summarizing current solutions for packaging defects. The results of the literature review highlight the lack of a general framework for dealing with problems in logistics packaging, which could provide clear guidance to companies on how to solve these problems. This includes a description of the types of errors in logistics packaging, their categorization, the method of detection, and an innovative framework for their elimination.

## 2. Systematic Literature Review

The literature search includes a literature analysis conducted using the systematic literature review (SLR) method. The systematic literature review methodology involves the searching, categorizing, selection, and analysis of relevant publications on a given topic. This analysis is carried out in a transparent manner, thus allowing for a comprehensive overview of the research topic under study. The research followed a set of steps (Figure 1) to achieve its objectives, ensuring a high-quality thesis and preventing the loss of scientific information. The aim of the literature analysis was to create a transparent scientific presentation of the topic under study while minimizing bias. The analysis includes a thorough search and analysis of extensive publications in English and Hungarian. A systematic literature search is a scientific study that does not require laboratory experiments, but does require prior planning and the rigorous application of the method [8–10].

The steps of a systematic literature analysis are as follows [8,11]:

1.  First, the motivation for the research has to established. The aim of the current research is to provide a deeper insight into the quality aspects of logistics packaging and to identify solutions that improve supply chain efficiency.
2.  Step 2 is to define the research questions.
    -   What scientific research has been published in this field since the 2000s?
    -   What research topics are being investigated?
    -   What are the current research limitations?
3.  The main aim of this step is to define keywords and their combinations. Academic papers were selected using "AND" combinations. In case of the current research, the keywords are "logistics", "packaging", "quality defects", and "digitalization".
4.  In the fourth step, the actual literature analysis is carried out. Four databases that support academic research were used in the current research. Of these databases, the search platforms of Scopus, ScienceDirect, and Web of Science were used throughout the entire analysis, while Google Scholar was also used excluding the last step. The time interval at the start of the analysis was defined as 2000 to 2023, which is shown in Figure 2.
5.  In the fifth step, the relevant literature has to be selected based on the literature review. After moderation of the results, and the selection and reading of relevant publications, the focus is on defining the main research direction. During the systematic literature

search, fields are narrowed down based on subject area, research field, and language. In the current research, publications that met the following fields were selected: research article, review article, chapter of book, minireview. The rest were not selected, and duplication was eliminated. The results of the analysis are shown in Figure 2.

6. Following the systematic review, the relevant literature can be summarized in a table, which in this case is performed in Table 1.

7. After reading all the publications and abstracts for the search results, a scientific gap can be identified, as was performed in the current research.

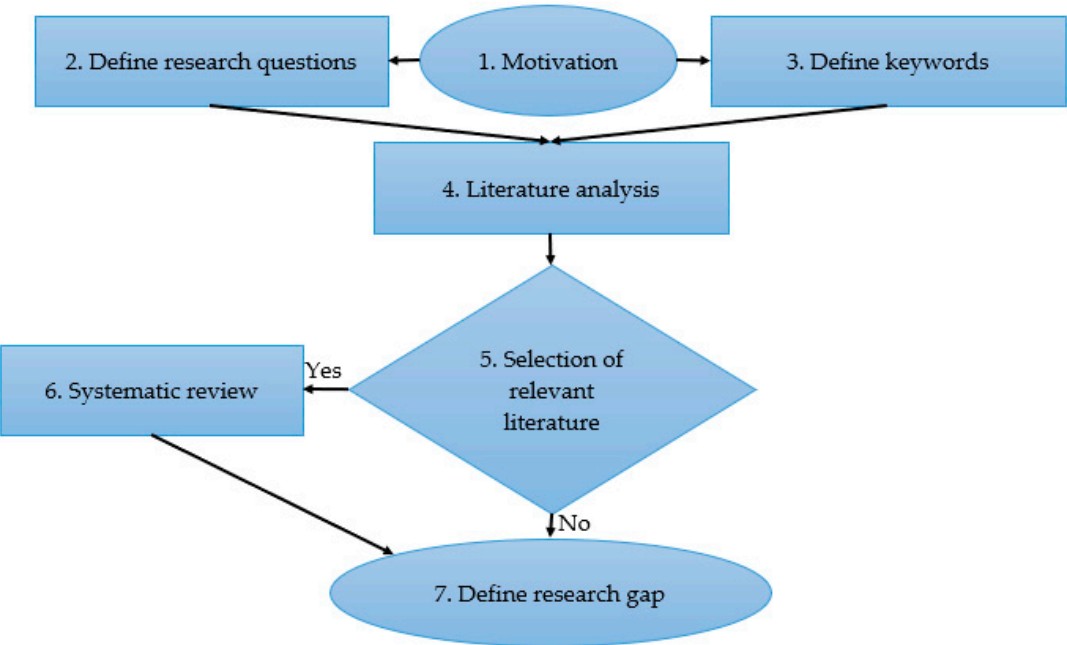

**Figure 1.** Steps of a systematic literature analysis (own editing).

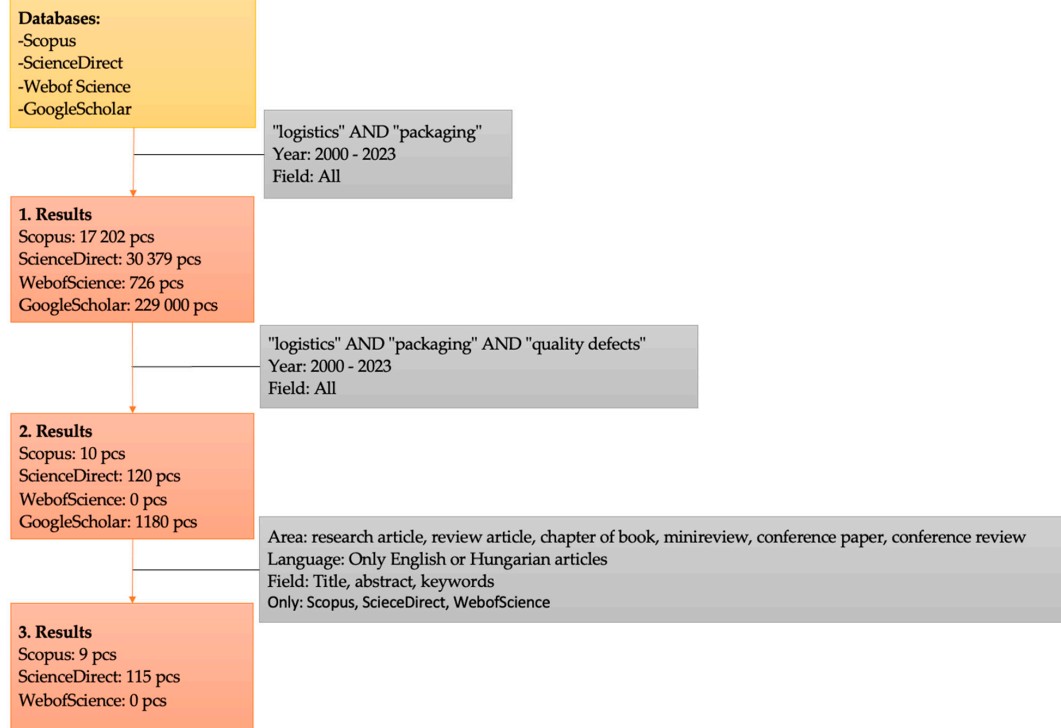

**Figure 2.** Results of literature analysis (own editing).

**Table 1.** Types of faults and methods for their detection and resolution (own editing).

| Fault Group | Fault Type | Method of Identification/Recognition | Problem Solving Method |
|---|---|---|---|
| A. Packaging order-system surface damage/deformation defect | A/1. External appearance, surface defect | Artificial vision, real-time fault detection system, camera [12–16] | Algorithm predicts errors [12] |
| | A/2. Dirty, contaminated packaging | Image acquisition system, camera, optical-based monitoring system [17–19] | Intelligent vision system [18] |
| | A/3. Holes, scratches | Electron microscope, cyclic voltammetry method [20–22] | Screen-printed carbon strips [21] |
| | A/4. Deformation error | Intelligent vision system [23] | - |
| B. Packaging system surface information damage, identification, traceability error | B/1. Bright printing, tearing | Visual inspection [24], machine vision technology [25] | Print industry development [25] |
| | B/2. Tracking problem | Recall effects, traceable resource units, comprehensive detail [26,27] | Building wireless networks [28], mixed integer linear programming method [27], |
| | B/3. Information damage | Not relevant information [16,29] | - |
| C. Error due to product placement | C/1. Fitting problem | Tall, long objects leave large empty spaces unpacked [30,31] | 3D packaging optimization [30], medical failure mode and effect analysis method [31] |
| | C/2. Packaging of irregularly shaped products | Irregular pieces, special shapes [32] | Constructive algorithm (HAPE) [32] |
| | C/3. Missing products in the packaging | Machine vision defect [33] | Shape template fitting algorithm [33] |
| D. Error due to product type | D/1. Perishable products | TTI (time and temperature indicator) [15,34–37] | Intelligent packaging [28,34,37], packaging with good tightness and closure properties [35], |
| | D/2. Weight problem | Weight measurement [38,39] | Electronic length measuring (LMD) increase sensitivity, replacement [38] |

*Results of the Systematic Literature Research*

Based on the content of the publications selected in the systematic literature search, Table 1 summarizes the errors found in packaging systems and the methods for their identification/recognition and handling.

Description of error groups and their recognition methods:

A: Packaging order-system surface damage/deformation defect: Defects that may cause surface damage or deformation defects belong to this group. This group was categorized into four subgroups.

- A/1: The YOLO (You Only Look Once) algorithm is a real-time system that specializes in evaluating images of packaging units for external defects and identifying defective or damaged products. The system can be used in factories and production lines, where it immediately assesses the quality of packaged boxes. The algorithm works in three main steps, which include object recognition, image analysis, and the sorting of defective products using robots [12]. By combining machine vision and learning, a surface defect detection method is applied, where the defective product is removed by a sorting robot system [13]. The main focus of the study [14] is on how blockchain technology supports the ecological integration of the coffee supply chain, and also addresses the identification of coffee packaging. Consumers can scan a QR code on

the packaging of the finished product to access the story behind it. Using an identifier with a self-supervising identifier (SSI), it allows producers to store and manage digital versions of identification documents, transaction receipts, or certificates in one place. Research [15] addresses the role of packaging in logistics, avoiding breakage, spoilage, and the hygiene of transported goods. The main objective of the study [16] is to identify and evaluate packaging elements by the food industry managers concerned. A questionnaire was used to conduct a survey, the results of which show that different companies have different perceptions of packaging shapes, colors, identification coding schemes, and consumption patterns.

- A/2: Dirty, contaminated packaging: Detection of logistic packaging units using SVM [17]. SVM, support vector machine-based error detection, is an image acquisition system that detects packaging errors. It provides real-time information about packaging errors in automatic sorting. The image acquisition process consists of a camera, optical lenses, and light sources. It is widely used for cartons [17]. The inspection of contaminated packaging can be performed by an optical-based inspection system. In this case, the packages, for example, ampoules, are illuminated and can be classified and detected in real time by the system [18]. Sterilization of products can be performed by conventional microbiological methods or by rapid microbiological methods [19].

- A/3: Holes, scratches: Punching and scratching of thin aluminum foil on a product is discussed in [20]. The causes of the formation of compounds on the surface and the effect on the quality of the final product are investigated by scanning electron microscopy (SEM) and energy dispersive spectroscopy (EDS). In [21], a method for the determination of the integrity of aseptic cardboard laminated packages containing aluminum foil by cyclic voltammetry was developed. The method is able to identify pinholes from cracks in the plastic inner layer. A study describes the suitability of cyclic voltammetry for the discrimination of defects in aseptic carton laminated packages containing aluminum foil [21]. One publication [22] deals with the detection of microcracks, voids, and layer breaks. It uses a C-mode scanning acoustic microscope (C SAM) tool for the nondestructive failure analysis of integrated circuit packages. It provides fast and comprehensive imaging of critical package defects and the location of these defects in three dimensions within the package [22].

- A/4: Deformation error: The paper [23] deals with the application of an intelligent vision system for the automatic inspection of two types of defects in glass products. For these applications, an installation consisting of a conveyor belt and a PC-controlled camera was used to simulate an industrial production line.

B: Packaging system surface information damage, identification, traceability error.

- B/1: Bright printing, tearing: For the bright print, tears are detected using the seven quality control methods. The lean quality control methods are check sheet, check diagram, cause and effect (Ishikawa) diagram, Pareto diagram, histogram, scatter diagram, flow chart. In the analysis, the packaging of granulated sugar was investigated using these methods [24]. The main objective of the paper [25] is to apply machine vision technology to the identification of surface defects in printed materials for outer packaging, which has resulted in reliable and efficient defect detection and can help the printing industry to develop. The method achieved a 99.4% detection rate and an average detection time of 103 ms for 2000 images, significantly improving the performance of the manual detection method [25].

- B/2: Tracking problem: The study [26] examines traceability issues in processed food. Traceability is challenging because resources are constantly changing. The paper presents various methods of traceability including physical separation, isotopic analysis, artificial intelligence, and blockchain-based approaches. The publication [27] deals with traceability systems and carbon emissions. The aim of the study is to develop a model for the integration of fish canning and distribution, considering traceability and carbon emissions to minimize the overall cost. The MILP (mixed integer linear programming) model is proposed to solve traceability problems.

- B/3: Information damage: Information damage, as well as the tracking problem, is also addressed in [26]. First, supply chain management deals with food traceability and its technology and information systems. Traceability is composed of three components: retrospective traceability, prospective traceability (these two categories belong to group B/2), and product history information. Product history information details the movement, time, identification of inputs, and operations that the product experiences in the supply chain [16,29].

C: Error due to product placement:

- C/1: Fitting problem: The research in [30] reviews the three-dimensional version of the classical NP-hard garbage packaging optimization problem, including its theoretical relevance and practical importance, and introduces new approximations based on previous research. The paper [31] explores the application implications of the medical error mode and the analysis of qualitative errors in device packaging. An analysis was conducted which revealed that the devices were not tested for functionality when packaged, and the device packaging was loose, which poses significant hidden risks to the overall device packaging process. The use of the medical failure mode and effects analysis method to control device packaging failures is effective in reducing the rate of quality failures [31].
- C/2: Packaging of irregularly shaped products: In [32], a set of irregularly shaped polyhedra is packed into a box-shaped fixed-size container, which is designed to accommodate all the polyhedra while minimizing loss. HAPE3D can deal with arbitrary shaped polyhedra, which can be rotated at different angles around each coordinate axis.
- C/3: The paper [33] discusses machine vision-based detection of medical packaging defects. The algorithm uses the sub-template fitting method to detect packaging defects in bubble wrap. The paper presents the software used for machine vision detection of pharmaceutical blister packaging.

D: Error due to product type.

- D/1: Perishable products: Packaging defects in perishable products can be detected by time and temperature indicators. It can record the thermal history and provide visual information when used in smart packaging. It can be used to record history and indicate the remaining expiry date [34]. The publication [35] analyzed the amount of raw material and waste processed over a defined period. It uses the SAP system for the analysis. The paper [28] also deals with perishable food packaging. It tracks the conditions affecting quality, temperature, and elapsed time. It uses sensors that monitor quality aspects. The research in [36] studies the permeability characteristics of different polymeric materials for different packaging materials under different environmental conditions. Temperature and humidity parameters are crucial for preserving food quality. The study [37] deals with smart packaging, which informs consumers about the freshness level of the packaged product without direct contact. The model dynamically updates the price of the packaged perishable product depending on its freshness, while reducing food waste and maximizing profit. The study in [15] also addresses breakage and is therefore included in the A/1 category, as well as in this category because of perishables.
- D/2: Weight problem: From a qualitative point of view, weight problems are addressed in [38] for yarn cone packaging. Electronic length gauges are used and a dimensional analysis is performed on the yarn length. Pareto diagrams were used to identify the main deviations and a Six Sigma method was applied. The research in [39] focuses on improving the weight inconsistency problem of coffee bean packaging. The research involved the definition, analysis, improvement, and verification of Six Sigma methodology and the application of related lean tools.

After completing the literature research, it was concluded that there is no framework that can effectively address the errors and problems found in the logistics packaging system.

### 3. Quality Assurance Framework for Logistics Packaging Systems

When reviewing the compliance of a packaging system used by a company, the procedure shown in Figure 3 applies to the established framework. The objective of the framework is to develop a concept that allows a qualitative solution to the packaging problems of logistics systems. The inductive method is used in this research, as the need to develop a packaging inspection framework has arisen through solving practical problems.

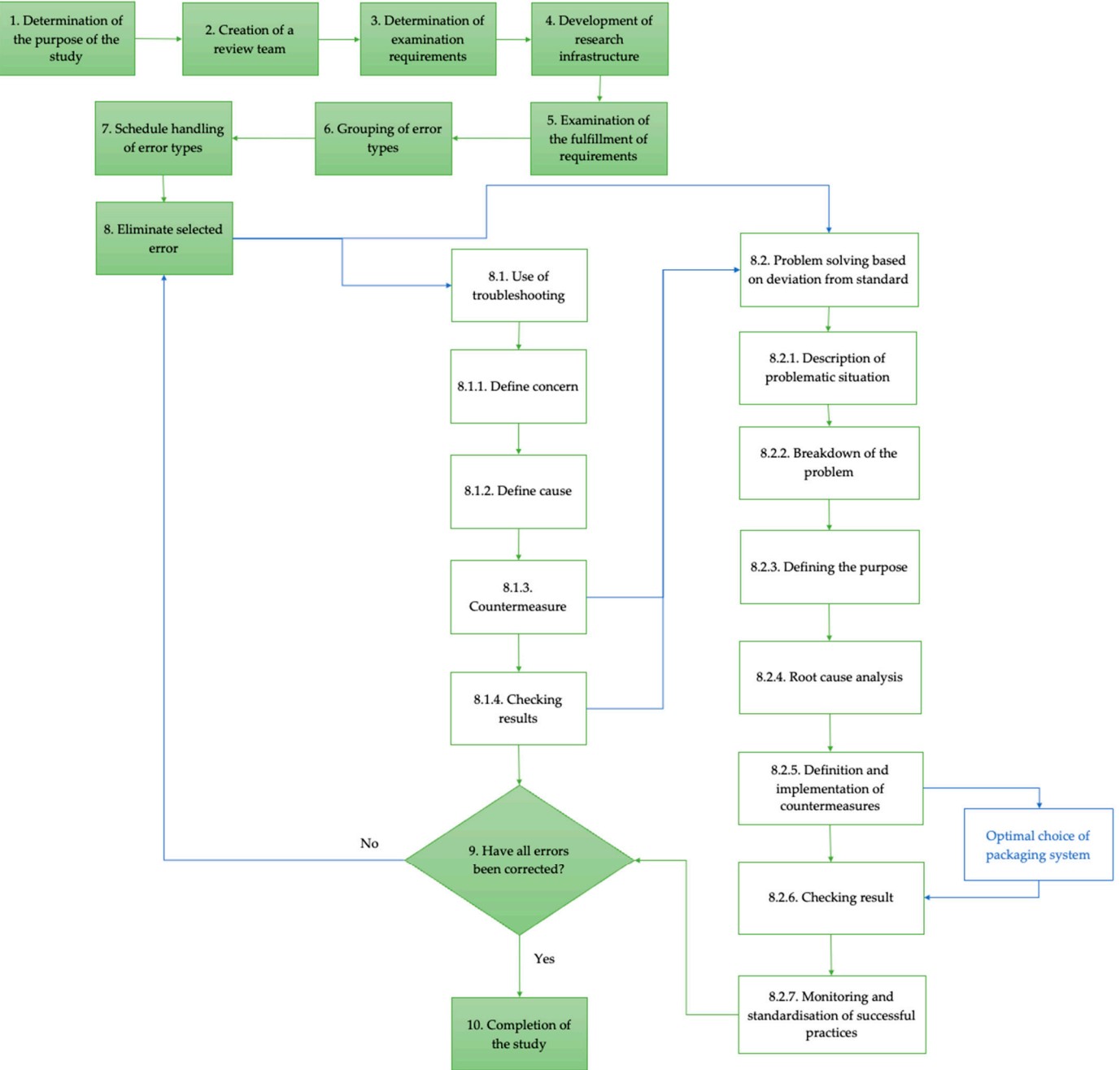

**Figure 3.** Packaging system compliance assessment framework (own editing).

1. Determination of the purpose of the study: The first step of the research is to identify the subsystem under examination for which the packaging system is going to be reviewed.
2. Creation of a review team: Depending on the nature of the subsystem under consideration, the group will be composed of internal and/or external actors. In the case of

tasks requiring specific expertise, the involvement of an external expert is justified, for instance in case of the packaging test and the simulation test.

3.  Determination of examination requirements: Two types of requirements can be distinguished, i.e., simple and complex testing requirements. Simple test requirements (verifiable by detection, measurement) are listed in Table 1. In this case, it is necessary to define which types of defects are relevant for the packaging system under examination and the methods to be used to detect them. The complex test requirements shall be fulfilled by means of packaging test tools and/or simulation modelling, which are based on [40]:

    -   Transit time: The transit time is the time between the objects bounding the delimited system. The transit time can be interpreted by the product type associated with the output points of the system under consideration and for the whole system [40].
    -   Total operating cost: The operating cost of a material flow system depends to a significant extent on the given packaging system and is determined for the system under consideration [40].
    -   Average usability in the process under consideration: This factor is used to consider the expected number of times the packaging system under consideration can be used [40].

4.  Development of research infrastructure: The systems necessary to test the fulfillment of the requirements defined in the previous point will be set up and, if necessary, an external expert will be involved, for example in case of the packaging test and the simulation test.

5.  Examination of the fulfillment of requirements: The given packaging is checked for compliance with the requirements of the inspection system, the results of which are summarized in Table 2.

**Table 2.** Template to define deviation from requirements (own editing).

|  | Fault Type | Method of Detection/Testing | Expected Value/Characteristic | Actual Value/Characteristic | Name of Deviation |
|---|---|---|---|---|---|
| Simple test requirements | Example: E/1. Deformation failure | Example: Intelligent vision system | Example: Maximum 10% deviation of the enclosure size of the heat treatment basket from the standard | Example: Maximum actual deviation 5% | Example: No discrepancy to check |
| Complex test requirements | Example: K/1. Transit time | Example: Analyzing tracking system data | Example: Average transit time of unit loads is less than 12.000 s | Example: Transit time 12.811 s/unit load | Example: 800 s/unit load |

6.  Grouping of error types: In cases where deviations from the requirements have occurred, errors have to be eliminated and problems have to be solved. The defects to be eliminated can be classified into groups based on the defects to be eliminated and the problem to be solved based on the deviation from the standard [40]. In the troubleshooting group, we classify those faults that are occasional; in other cases, the faults are classified in the other group.

7.  Schedule handling of error types: Based on the losses caused by the detected faults and the expected transit time for their elimination, it is necessary to prioritize and schedule the handling of the faults.

8.  Eliminate selected error: The problems identified will be solved in the order of the defined grouping and schedule. Step 8 is a cyclical process, as each cycle is used to eliminate one type of error.

    8.1.  Use of troubleshooting [41]: For failures in this category, the 4C method is used, with the following steps:

8.1.1.　Define concern: What do we know about the problem concerned?

8.1.2.　Define cause: What do we know about the cause?

8.1.3.　Countermeasure: What immediate or temporary countermeasures can we take to reduce the problem? Is a follow-up or more permanent countermeasure necessary to avoid recurrence? If so, the problem is dealt with in the other group.

8.1.4.　Checking results: Is everything in order? If the problem is repeated, the error will be moved to the other group.

8.2.　Problem solving based on deviation from standard [40]: In this case, the problem can be solved using the well-known seven steps of problem solving, which are:

8.2.1.　Description of problematic situation: Describing the current situation with facts and figures.

8.2.2.　Breakdown of the problem: Detailed breakdown of the problem by analysis, narrowing down.

8.2.3.　Defining the purpose: Determining how precisely and by when we want to reach the target.

8.2.4.　Root cause analysis: Explore cause and effect relationships, which must be verifiable.

8.2.5.　Definition and implementation of countermeasures: In a favorable case, one countermeasure can be defined for one problem. In this step, it is important to define the timeframe for implementation, communication, and ownership. In the case where an optimal choice of the packaging system to be used is required, the concept of a package-inspection system, as we have previously developed [39], can be applied.

8.2.6.　Checking result: Checking that the target has been reached. If the desired goal has not been reached, we go back to Step 6.2.4. or 6.2.5.

8.2.7.　Monitoring and standardization of successful practices: The countermeasure developed will be established as best practice and implemented in the long term. It is also important that, where possible, the practice developed can be shared with other areas and, if necessary, transferred.

9.　Have all errors been corrected? If there is still a packaging error to be corrected, the next error to be corrected is moved to Step 8, otherwise the test is completed.

10.　Completion of the study: If the errors detected are eliminated, the study is completed, and the experience gained should be analyzed and evaluated for future use.

## 4. Case Study on Handling Packaging Errors

1.　Determination of the purpose of the study: The case study is a review of the packaging system of a fictitious heat treatment plant, based on practical experience.

2.　Creation of a review team: The review team is made up of experts delegated by the sub-areas (production, logistics, maintenance, packaging management) and an external expert capable of carrying out the simulation test.

3.　For the system under consideration, the simple and complex test requirements have been defined as shown in Table 3 below.

4.　Development of research infrastructure: To check the fulfillment of the requirements defined in the previous point, it is necessary to develop a system capable of recognizing E/1 and E/3 errors. The system components needed to determine the other errors are already available.

5.　Examination of the fulfillment of requirements: Compliance with the requirements for a given packaging test system is checked and the results are summarized in Table 4.

**Table 3.** Definition of testing requirements (own editing).

| Test Requirements | Fault Type | Method of Definition/Recognition | Expected Value/Characteristic |
|---|---|---|---|
| Simple test requirement | E/1. Deformation failure | Intelligent vision system | Maximum 10% deviation of the enclosure size of the heat treatment basket from the standard |
| | E/2. Tracking problem | Unsuccessful reading, visual inspection | Heat treatment damages the unique identifiers on the basket |
| | E/3. Missing products in the packaging | Weight measurement | Not the required amount of product is placed in the heat treatment basket. The weight of the products in the basket should be between 290 and 300 kg. |
| Complex testing requirement | K/1. Transit time | Tracking system data analysis | Average transit time of unit loads is less than 12.000 s |
| | K/2. Average usability within the process tested | Tracking system data analysis | A unit load device must be used in at least 20 work cycles. |

**Table 4.** Identification of deviations from requirements (own editing).

| Examination Requirements | Fault Type | Method of Determination/Recognition | Expected Value/Characteristic | Actual Value/Characteristic | Name of Deviation |
|---|---|---|---|---|---|
| A simple test requirement | E/1. Deformation failure | Intelligent vision system | Maximum 10% deviation of the enclosure size of the heat treatment basket from the standard | Maximum actual deviation 5% | No discrepancy to check |
| | E/2. Tracking problem | Unsuccessful reading, visual inspection | Heat treatment damages the unique identifiers on the basket | The problem occurred in 1% of the cases studied | 1% of cases examined |
| | E/3. Missing products in the packaging | Weight measurement | The required amount of product is not placed in the heat treatment basket. The weight of the products in the basket should be between 290 and 300 kg. | The problem occurred in 1% of the cases studied | 2% of cases examined |
| Complex testing requirements | K/1. Transit time | Analyzing tracking system data | Average transit time of unit loads is less than 12.000 s | Transit time 12.811 s/unit load | 800 s/unit load |
| | K/2. Average usability within the examined process | Tracking system data analysis | A unit load device must be used in at least 20 work cycles. | 18 work cycle/unit document preparation tool | 2 work cycles |

6. Grouping of error types: As a result of the assessment, the packaging system tested failed two simple and two complex test requirements. Since the treatment of these failures cannot be combined, no grouping is possible.

7. Schedule handling of error types: Based on the losses caused by the detected defects and the expected transit time for their elimination, it is necessary to prioritize and schedule the handling of the defects from the beginning of the projects as shown in Table 5.

**Table 5.** Schedule for handling fault types (own editing).

| Fault Type | Problem Solving Period | Expected Result |
|---|---|---|
| E/2. Tracking problem | 1–4 weeks | Problem solving |
| E/3. Missing products in the packaging | 1–8 weeks | Problem solving |
| K/1. Transit time | 1–8 weeks | Problem solving |
| K/2. Average usability within the process tested | 1–8 weeks | Problem solving |

8.   Eliminate selected error: Managing simple requirements compliance in the short term requires the application of the 8.1 problem solving process, followed by the application of 8.2. The application of 8.1 avoids collateral losses due to failure, for instance, customer complaints; and the application of 8.2 eliminates them in the long term. Meeting complex requirements requires the application of Process 8.2. These problem-solving processes are best practices and will not be described in detail. The results achieved are summarized in Table 6.

**Table 6.** Eliminate selected errors (own editing).

| Fault Type | Problem Solution Result |
|---|---|
| E/2. Tracking problem | Process 8.1 resulted in the timely detection and replacement of non-identifiable unit load devices. Process 8.2 resulted in the application of a label with a higher resistance and the problem has been solved. |
| E/3. Missing products in the packaging | As a result of Process 8.1, unit loads with inappropriate content were identified by weighing, and the problem was temporarily solved after the content was corrected. Process 8.2 resulted in a new filling system to ensure that the correct quantity of the product is placed. |
| K/1. Transit time | As a result of Process 8.2, the transit time could be ensured by choosing a new unit load device, as its handling time was found to be shorter. The selection of the ideal unit load device required the application of a simulation test method [39]. |
| K/2. Average usability within the process tested | Testing of the new unit-loading tool selected as a result of the K/1 problem-solving exercise has shown that it meets the requirements. |

9.   Have all errors been corrected?: All types of errors identified during the investigation have been corrected and the investigation is now complete.
10.   Conclusion of the study: The potential applications of the solutions developed in the study in other areas have been explored.

Based on the literature review, the solutions applied in the field thus far were compared and presented. The methods developed are isolated and a framework for quality assurance in packaging systems has not been developed. In addition, it is important to mention the importance of the training of people working within the company, with regard to the application of the developed procedure, for which the development and introduction of training materials is planned.

**5. Conclusions**

The review of the packaging systems used by companies is becoming increasingly necessary due to the frequently changing product portfolio and the dynamically changing environmental impacts. As a result of these changes, the requirements for the used packaging systems are often not met and many operational errors occur. This generates clear losses in a wide variety of areas, for instance in the case of storage, transport, manufacturing, etc.; the effective elimination of which is a relevant factor in determining the competitiveness of companies. Research experience in companies and the results of a systematic literature review show that there is no general method for dealing with packaging problems, and that companies use isolated solutions, which typically increase the time needed to solve

the problem and may lead to important investigative aspects being overlooked. In the development of the concept, we identified and systematized packaging errors in the literature and their detection and resolution methods, and developed a general framework for detecting and eliminating packaging errors. The validity of the methodology was tested and validated through a case study. The framework is currently suitable for quality assurance of packaging systems for homogeneous unit loads and is planned to be extended to inhomogeneous unit loads in the future. Further development of the developed framework is planned based on the practical experience gained from the expected application.

**Author Contributions:** Conceptualization, H.M. and P.T.; investigation, H.M. and P.T.; methodology, H.M. and P.T.; supervision, P.T.; writing—original draft, H.M.; writing—review and editing, H.M. and P.T. All authors have read and agreed to the published version of the manuscript.

**Funding:** This research received no external funding.

**Data Availability Statement:** Data are contained within the article.

**Conflicts of Interest:** The authors declare no conflict of interest.

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
