# Peer review of "An Innovative Framework for Quality Assurance in Logistics Packaging"

_logistics, 2023_

Round 1

Reviewer 1 Report

Comments and Suggestions for Authors

It is obvious that much time and efforts are put into this research work, and I appreciate it. However, I have some observations, as follow.

1. The abstract needs to be improved. There are too many words to introduce the research background, but there is a lack of explanation of the research methods, research conclusions, and research contributions.

2. The keywords fail to highlight the main research question of this paper.

3. In section 2, the literature review is too long and lacks structural logic.

4. In section 3, there is a shortage of rigorous academic research methods.

5. The conclusions are too short and needs to highlight the contribution of research innovations and strengthen the explanations of future research.

6. Make a grammar check. There are some minor problems.

Comments on the Quality of English Language

Minor editing of English language.

Author Response

Dear Reviewer!

Thank you for the valuable suggestions, based on which we modified the attached paper. The corrections have been marked with proofreading for verifiability.

We made the following changes:
- The abstract and the summary section have been significantly modified.
- The keywords have been modified.
- We reviewed and modified the figures. Their quality has been improved.
- We clarified and, if necessary, supplemented the content descriptions.
- We corrected the literature review in order to verify the new content.
- In order to significantly improve the quality of English grammar, we reviewed the paper with several colleagues, and made corrections based on their comments.

We trust that, based on the modifications made, the thesis meets the expectations.

Best regards,

The authors

Reviewer 2 Report

Comments and Suggestions for Authors

The objective of this work was to present the concept of a general framework for managing packaging defects aligned with the company's requirements.

In my opinion, the description of the problem is clear, but what the effective contribution of this work was is not clear. My suggestion would be to rewrite the conclusions, clearly demonstrating the extent to which this work is an important contribution to for quality assurance in logistics packaging.

 Figures 2 and 3 should be improved as they are difficult to read.

Comments on the Quality of English Language

It is readable.

Author Response

(The authors gave the same response as above.)

Reviewer 3 Report

Comments and Suggestions for Authors

The paper topic is interesting but needs to be revised. As per the following points 

1. The motivation and novilities should written in a brief way. 

2. The data sources are not properly cited. 

3. There should be some relatable application of the proposed approach. 

4. The abstract should be written in six sub-parts a) What is the purpose? b) Research question c) methodology d) validation e) results f) why are your results significant? 

5. Please check all table numbers (like in line number 292 on page 9). 

6. All figures and tables should be cited in the text. 

7. There should be a comparative study on different studies. 

8. In conclusion, the manager inside should be added.

9. More remarks and note should be added.

Comments on the Quality of English Language

Extensive editing of English language required

Author Response

(The authors gave the same response as above.)

Round 2

Reviewer 1 Report

Comments and Suggestions for Authors

 Accept in present form

Comments on the Quality of English Language

Minor editing of English language required

Reviewer 2 Report

Comments and Suggestions for Authors

The paper is accept in present form

Comments on the Quality of English Language

Minor editing of English language

Reviewer 3 Report

Comments and Suggestions for Authors

The authors revise the paper very carefully. I recommend it for publication in current form.

Comments on the Quality of English Language

Extensive english editing required.